# High ferritin is associated with liver and bone marrow iron accumulation: Effects of 1-year deferoxamine treatment in hemodialysis-associated iron overload

Lucas L. A. Nunes[1]*, Luciene M. Dos Reis[1], Rosse Osorio[2], Hanna K. A. Guapyassú[1], Rosa M. A. Moysés[1], Hilton Leão Filho[3], Rosilene M. Elias[1,4], Carlos E. Rochitte[2], Vanda Jorgetti[1☉], Melani R. Custodio[1☉]

1 LIM 16 –Laboratorio de Fisiopatologia Renal, Hospital das Clinicas HCFMUSP, Faculdade de Medicina, Universidade de Sao Paulo, Sao Paulo, SP, Brazil, 2 Radiology Department, Hospital das Clinicas HCFMUSP, Universidade de Sao Paulo, Sao Paulo, Brazil, 3 Universidade Nove de Julho (UNINOVE), Sao Paulo, Brazil, 4 Radiology, Instituto do Coracao (InCor), Hospital das Clinicas HCFMUSP, Universidade de Sao Paulo, Sao Paulo, Brazil

☉ These authors contributed equally to this work.
* lucaslan@ufpa.br

## Abstract

### Background

Iron (Fe) supplementation is a critical component of anemia therapy for patients with chronic kidney disease (CKD). However, serum Fe, ferritin, and transferrin saturation, used to guide Fe replacement, are far from optimal, as they can be influenced by malnutrition and inflammation. Currently, there is a trend of increasing Fe supplementation to target high ferritin levels, although the long-term risk has been overlooked.

### Methods

We prospectively enrolled 28 patients with CKD on hemodialysis with high serum ferritin (> 1000 ng/ml) and tested the effects of 1-year deferoxamine treatment, accompanied by withdrawal of Fe administration, on laboratory parameters (Fe status, inflammatory and CKD-MBD markers), heart, liver, and iliac crest Fe deposition (quantitative magnetic resonance imaging [MRI]), and bone biopsy (histomorphometry and counting of the number of Fe positive cells in the bone marrow).

### Results

MRI parameters showed that none of the patients had heart iron overload, but they all presented iron overload in the liver and bone marrow, which was confirmed by bone histology. After therapy, ferritin levels decreased, although neither hemoglobin levels nor erythropoietin dose was changed. A significant decrease in hepcidin and FGF-23 levels was observed. Fe accumulation was improved in the liver and bone marrow, reaching normal values only in

**Data Availability Statement:** All relevant data are within the manuscript and its Supporting Information files.

**Funding:** This work was supported by the Brazilian National Council for Scientific and Technological Development (CNPq; grant number 434758/2018-3). The funders had no role in study design, data collection and analysis, decision to publish, or preparation of the manuscript. The funders had no role in study design, data collection and analysis, decision to publish, or preparation of the manuscript.

**Competing interests:** The authors have declared that no competing interests exist.

the bone marrow. No significant changes in turnover, mineralization or volume were observed.

## Conclusions

Our data suggest that treatment with deferoxamine was safe and could improve Fe accumulation, as measured by MRI and histomorphometry. Whether MRI is considered a standard tool for investigating bone marrow Fe accumulation requires further investigation.

**Registry and the registration number of clinical trial:** ReBEC (Registro Brasileiro de Ensaios Clinicos) under the identification RBR-3rnskcj available at: https://ensaiosclinicos.gov.br/pesquisador.

## Introduction

Anemia is a major problem among patients with chronic kidney disease (CKD), mainly in those on maintenance hemodialysis [1] factors such as absolute or functional iron (Fe) deficiency, relative erythropoietin deficiency, and a chronic inflammatory state. Several studies have shown that anemia worsens patients' quality of life, favors the development of cardiovascular complications, and increases the mortality rate. Over the years, there has been a reduction in blood transfusion dependence associated with an increase in Fe replacement and the use of erythropoiesis-stimulating agents [2, 3].

Serum Fe, ferritin, and transferrin saturation (TSAT) levels are used in clinical practice to guide Fe supplementation. However, these parameters are far from optimal, as they can be influenced by malnutrition and inflammation, both of which are common in patients with CKD. The KDIGO (Kidney Disease: Improving Global Outcomes) guidelines suggest Fe supplementation if ferritin levels are <500 ng/dL or TSAT is < 30% [4]. However, Rostoker G et al. [5] using quantitative magnetic resonance imaging (MRI) to document hepatic hemosiderosis in 84% of patients undergoing hemodialysis (HD), and in 30% of patients, the burden was as intense as that observed in patients with genetic hemochromatosis. Notably, the median serum ferritin level was 446 ng/ml, which is below the current threshold that demands discontinuation of intravenous Fe administration.

Liver MRI is considered the gold standard noninvasive method for estimating and monitoring iron stores in secondary and genetic hemosiderosis, but MRI is not often performed in patients with CKD. Iron overload can lead to heart failure, a complication frequently seen in patients with thalassemia and in those with hereditary hemochromatosis [5, 6] In patients with CKD, heart failure is frequent, although the contribution of iron overload is little appreciated. Another site for Fe accumulation that has been evaluated over the last decade is the bone marrow. Bone biopsies performed for different indications have shown Fe deposits in the mineralization front and bone marrow of these patients [7–9].

Deferoxamine (DFO), an iron binder often used by thalassemia patients, is capable of increasing the levels of hemoglobin and decreasing the levels of ferritin and tissue Fe [10, 11]. In CKD, DFO has been widely used as a binder in patients with aluminum (Al) intoxication [12]. However, few studies have reported the use of DFO to treat iron overload in patients with CKD, mostly in the pre-ESA era [13–15].

Currently, there is no consensus on the optimal ferritin range, how to evaluate Fe deposition, or whether DFO can be used to treat iron overload in patients with CKD. In the present study, we prospectively enrolled patients with CKD undergoing hemodialysis with serum

ferritin > 1000 ng/ml, and, we tested the effects of 1-year DFO use, on laboratory parameters, bone biopsy, myocardial, liver, and bone Fe accumulation, assessed by MRI by T2* values, R2* relaxometry and the use of the R2* Water parameter, which improves the diagnostic accuracy of iron overload.

## Patients and methods

With the informed written patients' consent and local ethics committee (Hospital das Clinicas da Faculdade de Medicina da USP) approval (CAPpesq # 1.906.167), 28 patients from São Paulo/SP, Brazil, undergoing chronic intermittent HD were enrolled in this prospective study during a 12-month period from February 3$^{st}$,2017, after approval by the Ethics Committee, to February 28th, 2019, for participant recruitment (February 2017 to February 2018) and follow-up (1 year). The sample size was obtained by convenience. The main outcome was organ iron deposition measured by magnetic resonance imaging as R2*Water.

The inclusion criteria were ferritin levels >1,000 ng/ml, age > 18 years, on an HD schedule of 3 times/week for at least six months. The exclusion criteria were refusal to participate in the study, ethanol or drug abuse, claustrophobia, hepatic cirrhosis, active malignancy, HIV infection, hepatitis B and C, current use of steroids, presence of cardiac pacemakers or metallic cardiac valves, previous kidney transplant, and previous DFO treatment.

This study was registered at the REBEC (Registro Brasileiro de Ensaios Clinicos) at the website https://ensaiosclinicos.gov.br/welcome under the identification number #RBR-3rnskcj (Recruitment—Feb 2017 to Feb 2019 and Follow-up—1 year).

The clinical and demographic parameters included age, sex, race, primary cause of CKD, HD duration, and vascular access. Fasting blood samples were obtained on the same day as the MRI examination. The samples were centrifuged, aliquoted in cryovials, and stored at -80˚C. Serum levels of hemoglobin (Hb) Fe, transferrin saturation ferritin, total calcium (tCa), ionized calcium (iCa), phosphate, total alkaline phosphatase (AP), and C-reactive protein (CRP) (immunoturbidimetric method), were analyzed using standard laboratory techniques. Intact PTH (Immulite; DPC-Biermann, Bad Nauheim, Germany,), 25(OH) vitamin D (immunoassay), intact FGF 23 (ELISA FGF 23 Human intact-Quidel San Diego, CA, USA), C-terminal FGF-23 (Human FGF-23 C-Term, ELISA, Quidel San Diego, CA, USA,), and Human Hepcidin (Quantikine ELISA, kit R&D Systems Inc, Minneapolis, MN, USA,) were used according to the manufacturer's instructions.

Patients underwent liver, heart, and bone (iliac and lumbar spine) MRI and iliac crest bone biopsy at the beginning and after 12 months of DFO administration (5 mg/kg once a week during the last hour of the second HD session of the week). The patients did not receive blood transfusions or iron infusions during the study period (S1 File).

### Bone biopsy and histomorphometry, magnetic resonance imaging and statistical analysis

Details of the bone biopsy, histomorphometry, MRI protocol and Statistical analysis are given in S2 File.

## Results

### Baseline

Of the 70 patients selected, 28 were included in this study (Fig 1) The baseline demographic, clinical, and laboratory features and histomorphometry data are shown in S1 and S2 Tables. Most patients were non-white men, relatively young, and on dialysis for a median time of 36

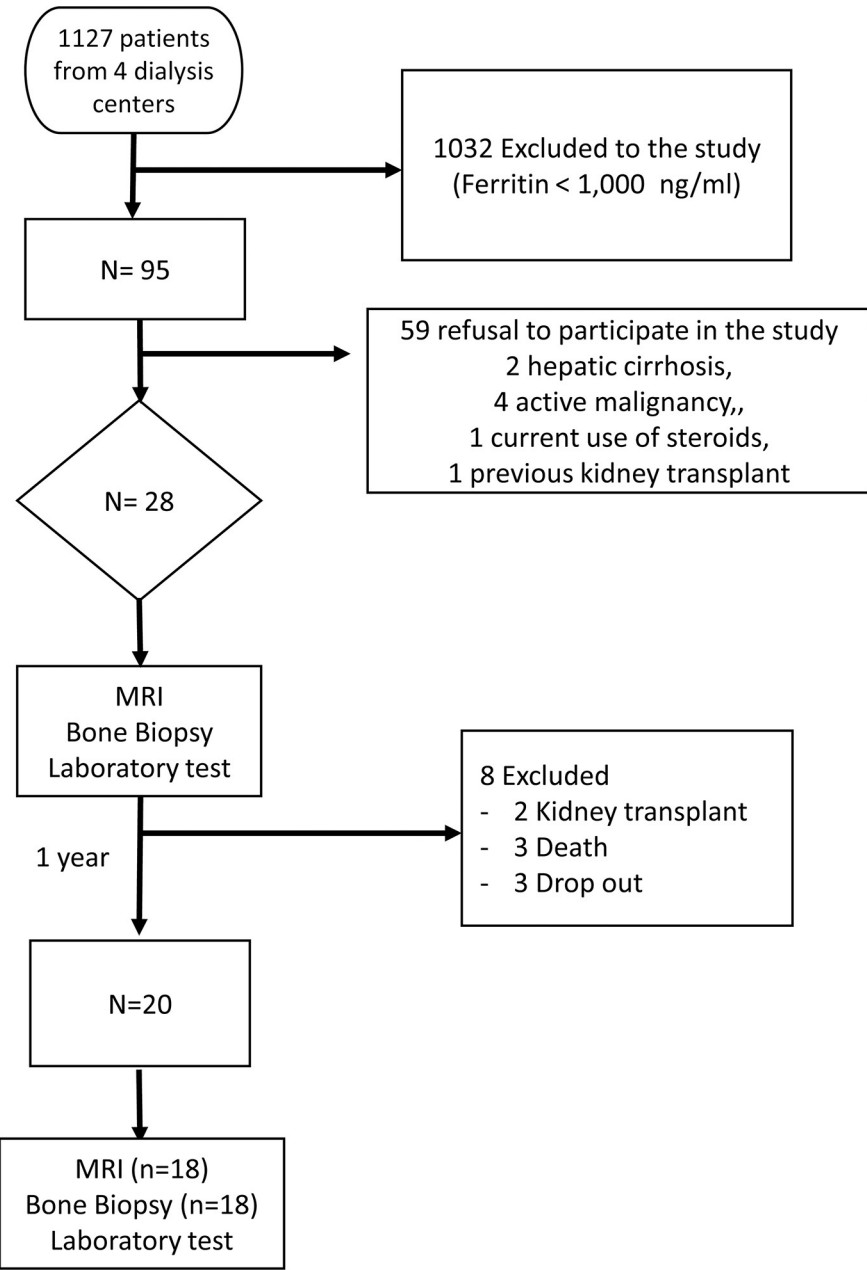

**Fig 1. Patient flow diagram.** Abbreviations: MRI = Magnetic Resonance imaging.

months. Diabetes and hypertension account for > 60% of the etiology of CKD. The median weekly EPO and monthly iron sucrose doses in the previous year were 160 UI/kg and 258 mg, respectively. C-reactive protein levels increased in 50% of the patients.

MRI parameters (S3 Table) showed that none of the patients had iron overload in the heart, but all of them had iron overload in the liver. Using the T2* parameter [16–18], mild and moderate Fe depositions were found in 22 (78.6%) and six (22.4%) patients, respectively. All patients showed liver Fe accumulation using the R2*Water parameter. Ferritin correlated with Liver R2*Water (Fig 2A) and LIC (r = 0.735, p < 0.0001). No correlation was found with any other laboratory parameter. Regarding the lumbar spine, we found that 23 of 28 patients (82.1%) had MRI

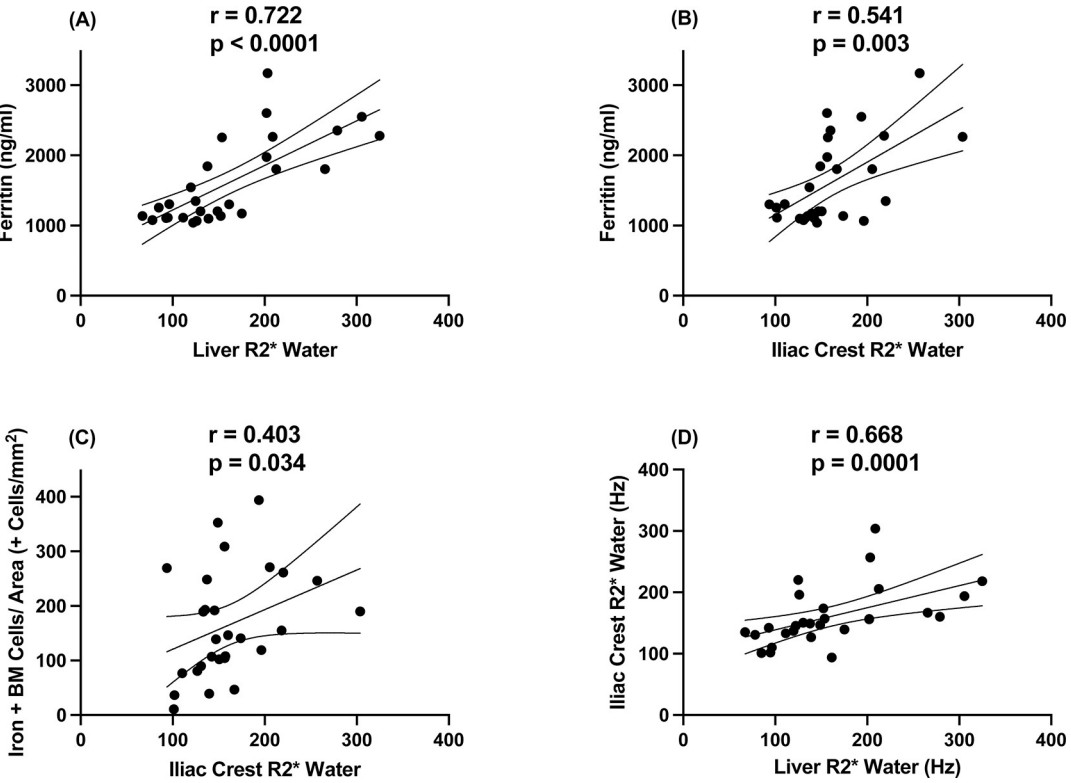

**Fig 2. Univariate correlations between serum ferritin, magnetic resonance imaging (MRI), and bone biopsy parameters.**
A. Serum ferritin and Liver R2*Water. B. Serum ferritin and right iliac crest R2*Water. C. Right iliac crest R2*Water and number of Fe+ bone marrow cells adjusted by bone marrow area. D. Right iliac crest R2*Water and liver R2*Water.

signs of iron overload. There was a positive correlation between the lumbar spine R2*Water and monthly doses of intravenous iron (r = 0.440; p = 0.019), but there was no correlation between Spine R2*water with Liver R2* water (r = -0.158, p = 0.422) and no correlation between serum ferritin and spine R2*water (r = -0.023, p = 0.907) at baseline. In addition, 18 (64.3%) and 17 (60.7%) patients showed Fe accumulation in the iliac crest using R2* Water and R2*, respectively.

Right iliac crest R2*Water correlated with serum ferritin and the number of Fe+ bone marrow cells adjusted by bone marrow area (Cells Fe+/Ma.Ar), as shown in Fig 2B and 2C, respectively. Similar results were obtained in the left iliac crest. R2*Water in the liver correlated with R2*Water at the right iliac crest (Fig 2D), but not at the lumbar spine (r = 0.015; p = 0.938).

According to the TMV classification, (Turnover, Mineralization, Volume) aimed to designate the quantity (volume) and the quality (turnover and mineralization) of the bone, patients were distributed between low (33.3%) and high (66.7%) turnover. Bone mineralization was normal in 33.3% of patients and abnormal in 66.7%. The volume was normal in 66.7%, low in 27.7%, and high in the remaining 5.6% of the patients. Of note, patients with low bone turnover had a higher monthly Fe dose in the previous year than those with high turnover (416.5 ± 147.4 vs. 240.1 ± 161.8 mg/month, respectively: p = 0.04). However, Fe was not detected at the mineralization front in any of the patients.

## Follow-up

From the original cohort of 28 patients, 2 underwent a kidney transplant, 3 died (2 due to cardiovascular events and 1 due to respiratory infection), and 3 withdrew their consent.

**Table 1. Laboratory parameters and bone mineral density before and after 12 months after DFO.**

|  | Reference range | Baseline | 12 months | Absolute change | p |
|---|---|---|---|---|---|
| Hemoglobin (g/dl) | 13.5–17.5 g/dL | 11.3 ± 2.1 | 11.3 ± 1.4 | 0.01 ± 0.6 | 0.974 |
| Iron (ug/dl) | 65–175 μg/dL | 109.3 ± 44.0 | 66.6 ± 26.5 | -42.7 ± 8.7 | **0.001** |
| Transferrin saturation (%) | 20–40% | 48.8 ± 20.6 | 28.6 ± 11.0 | -20.2 ± 5.2 | **0.0001** |
| Ferritin (ng/ml) | 13–150 ng/mL | 1,279 (1,112–2,197) | 441 (272–1,169) | -832.7 ± 165.6 | **0.001** |
| Total calcium (mg/dl) | 8.5–10.5 mg/dL | 9.6 ± 0.6 | 9.2 ± 0.7 | -0.4 ± 0.12 | **0.004** |
| Ionized calcium (mg/dl) | 4.6–5.3 mg/dL | 4.97 ± 0.27 | 4.80 ± 0.39 | -0.17 ± 0.08 | **0.034** |
| Phosphate (mg/dl) | 2.3–4.7 mg/dL | 5.4 ± 1.7 | 4.6 ± 1.5 | -0.8 ± 0.46 | 0.121 |
| Alkaline phosphatase (UI/l) | 35–104 U/L (F) 40–129 U/L (M) | 135 (99–217) | 188 (124–286) | 0.02 ± 0.61 | **0.002** |
| PTH (pg/ml) | 10–65 pg/mL | 283 (176–980) | 451 (224–900) | 82.4 ± 22.1 | 0.296 |
| 25(OH)D (ng/ml) | 30–100 ng/ml | 27.1 ± 12.1 | 27.6 ± 8.4 | 0.5 ± 2.7 | 0.872 |
| C-reactive protein (mg/dl) | <0.5 mg/dL | 4.9 (3.0–9.7) | 5.0 (3.0–11.0) | 0.2 ± 2.1 | 0.856 |
| FGF23i (pg/ml) | 18–73 pg/ml | 1,443 (492–5,766) | 504 (225–1,791) | -1,429 ± 646 | 0.112 |
| FGF23c (RU/ml) | 21.6–91.0 RU/L | 1,231 (832–7,240) | 1,057 (564–3,100) | -1,833 ± 772 | **0.002** |
| Hepcidin (ng/ml) | Not determined | 181.8 ± 37.6 | 137.6 ± 53.9 | -44.2 ± 15.0 | **0.009** |
| Erythropoietin (UI/kg/week) | - | 44 (23–49) | 41 (28–51) | 0.45 ± 4.4 | 0.999 |
| Calcitriol (μg/week) | - | 0 (0–0.4) | 0 (0–0.5) | 0.05 ± 0.03 | 0.500 |
| Cinacalcet (mg/day) | - | 0 (0–0) | 0 (0–0) | 0 | 0.125 |
| Sevelamer (g/day) | - | 4.8 (4.8–7.2) | 4.8 (2.4–7.2) | -0.48 ± 0.41 | 0.406 |

Values are mean ± SD or median (25,75) for variables at baseline and 12 months and mean ± SEM for absolute change. Bold p values are < 0.05. 25(OH)D, 25-hydroxy-vitamin D; PTH, parathyroid hormone; FGF23i, intact fibroblast growth factor 23; FGF23c, carboxy-terminal fibroblast growth factor; (F): Females; (M): Males

Therefore, we analyzed a final sample of 20 patients (13 men, 56 ± 12 years). None of the patients required blood transfusion or experienced adverse effects related to DFO administration. Table 1 shows the changes in laboratory parameters during the follow-up. There was a decrease in serum Fe, transferrin saturation, and ferritin, although 9 patients (45%) remained with a ferritin level higher than that prescribed by the KDIGO guidelines [4] (> 500 ng/ml) at the end of follow-up. These patients did not have significant differences in baseline ferritin (1,871.7 ± 677.6 vs. 1,410.9 ± 529.4; p = 0.105). However, liver R2*Water was higher in these patients at baseline (207.2 ± 74.2 vs. 125.6 ± 38.0; p = 0.005), as well as at the end of the follow-up (175.6 ± 56.2 vs. 71.9 ± 25.1; p < 0.001), indicating that they had more accumulation of Fe. For the entire group, we found a significant decrease in C-terminal FGF-23 and in Hepcidin levels. No significant changes were found in the erythropoietin doses. A decrease in the serum calcium level was not accompanied by a significant increase in PTH. There were no significant changes in calcitriol, cinacalcet, or sevelamer dose. Dialysis prescription was not modified during the study.

There was a significant reduction in Fe deposition in the liver, lumbar spine, and iliac crest (Table 2 and Fig 3). All but 2 patients showed improved liver R2*Water.

However, none of them reached normal R2*Water values during follow-up (Fig 4A). Regarding the iliac crest, all patients showed improvement and had normal R2*Water values (Fig 4B).

A significant decrease in the number of Fe+ bone marrow cells was observed. Bone histomorphometry revealed an increase in trabecular separation and a decrease in the osteoblastic surface (Table 3). There were no significant changes in turnover (p = 0.093), mineralization (p = 1), or volume (p = 1) after 1 year (Table 4 and Fig 5).

**Table 2. Magnetic resonance imaging (MRI) of liver, heart, lumbar spine, and iliac crest at baseline and at the end of follow-up.**

| | Baseline | 12 months | Absolute change | p |
|---|---|---|---|---|
| **Liver** | | | | |
| LIC (mg/g) | 4.35 ± 2.15 | 3.08 ± 2.14 | -1.27 ± 0.27 | **<0.001** |
| LIC (µmol/g) | 79.31 ± 40.63 | 55.07 ± 38.23 | -24.24 ± 4.86 | **<0.001** |
| T2* (mS) | 6.90 ± 2.93 | 10.92 ± 6.18 | 4.02 ± 1.04 | **0.001** |
| R2* Water (Hz) | 159.69 ± 65.51 | 117.96 ± 66.70 | -41.73 ± 8.78 | **<0.001** |
| R2* (Hz) | 171.49 ± 76.46 | 126.14 ± 76.12 | -45.35 ± 9.70 | **<0.001** |
| **Lumbar spine (L3)** | | | | |
| T2* (mS) | 8.5 (7.5–11.0) | 11.1 (9.1–14.3) | 0.36 ± 2.93 | **0.042** |
| R2* Water (Hz) | 173.74 ± 40.48 | 123.07 ± 41.70 | -50.67 ± 12.63 | **<0.001** |
| R2* (Hz) | 113.11 ± 32.45 | 93.73 ± 26.69 | -19.38 ± 7.91 | **0.025** |
| **Right iliac crest** | | | | |
| T2* (mS) | 9.03 ± 2.15 | 11.59 ± 3.00 | 2.56 ± 0.87 | **0.006** |
| R2* Water (Hz) | 156.28 ± 42.47 | 125.81 ± 52.01 | -30.47 ± 9.10 | **0.004** |
| R2* (Hz) | 116.57 ± 25.95 | 90.26 ± 25.25 | -26.31 ± 6.99 | **0.002** |
| **Left iliac crest** | | | | |
| T2* (mS) | 9.14 ± 2.44 | 12.56 ± 3.48 | 3.42 ± 0.87 | **0.004** |
| R2* Water (Hz) | 154.65 ± 43.43 | 119.25 ± 52.43 | -35.4 ± 10.2 | **0.003** |
| R2* (Hz) | 114.71 ± 28.49 | 86.41 ± 27.75 | -28.3 ± 9.67 | **0.001** |

Values are mean ± SD or median (25,75) for variables at baseline and 12 months and mean ± SEM for absolute change. Bold p values are< 0,05; ms: miliseucounds; Hz: Hertz; mg/g: milligram/gram; LIC: liver iron concentration; R: right; L: left; µmol/g: micromole/gram

## Discussion

The present study reports three main findings. First, patients on maintenance HD with high serum ferritin present iron overload in the liver and bone marrow, but not in the heart. Second, the results were obtained using the MRI R2*Water parameter, which excludes fat measurement, particularly in the bone marrow, which showed a high-fat fraction in the analysis. Third, Fe overload improved after stopping Fe supplementation and the use of DFO, leading to a reduction in serum ferritin, transferrin saturation index, serum Fe, FGF23c and Hepcidin levels, and an improvement in liver and bone marrow MRI findings. These results were detected despite no changes in serum hemoglobin or erythropoietin doses.

Iron overload was described in patients with CKD before the introduction of erythropoietin due to frequent blood transfusions to correct anemia [19]. However, after the EPO era, blood transfusions were reduced. Therefore, the administration of intravenous iron in association with erythropoietin has become a common clinical practice to correct anemia in patients on dialysis. Hereafter, it is expected that supplemental Fe is harmless. However, this concept was challenged by the study by Rostoker et al. [5], which evaluated patients on HD treated with EPO and regular Fe supplementation. Severe Fe overload was found in more than one-third of patients, indicating the need for more accurate monitoring of Fe supplementation in this population. Serial hepatic MRI showed a clear relationship between iron therapy and increased hepatic iron stocks, confirming the deleterious role of intravenous iron therapy [5]. Another study from the same group showed that intravenous Fe administration should not exceed 250 mg/month, a cutoff above which represents a 3.9 times higher chance of Fe liver accumulation, evaluated through MRI [20].

Before the introduction of EPO, Ali *et al* described that there was a discrepancy regarding iron overload in patients with CKD, i.e., there was iron overload in the liver tissue, but not in

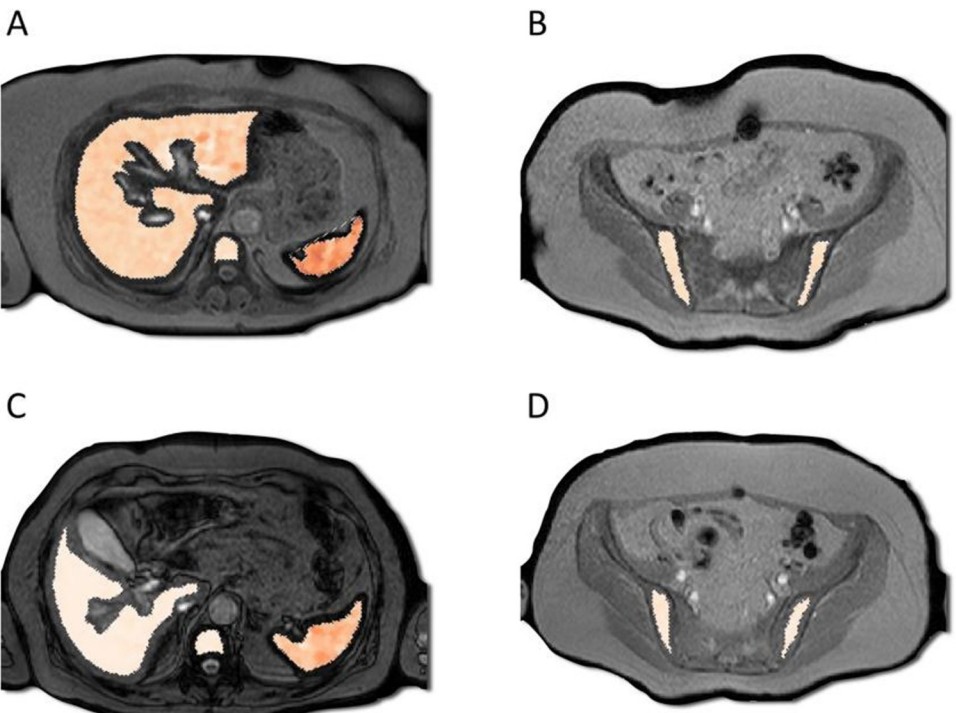

**Fig 3. Illustrative MRI of a given patient at baseline and at the end of follow-up.** The figure shows R2*Water maps (R2*Water) overlap in the axial images of the upper abdomen and pelvis before (A and B) and after (C and D) treatment. The liver R2*Water showed a median value of 207 s$^{-1}$ in A and 81 s$^{-1}$ in C after treatment, which correlated with the reduction in serum ferritin levels. The same can be observed in the spleen, from 373 s$^{-1}$ in A to 258 s$^{-1}$ in C. The bones also demonstrated a reduction in the R2*Water values, ranging from 148 s$^{-1}$ to 94 s$^{-1}$ for the L3 vertebra (A and C) and from 151 s$^{-1}$ to 95 s$^{-1}$ (B and D) in the left iliac crest. Republished from Body Digital Pte Ltd under a CC BY license, with permission from Wanida Chua-anusorn, original copyright 2024.

the bone marrow [21]. Contradictorily, these results were not the same as observed in the studies by Rostoker *et al. [22]*, Carrilho *et al. [23]*, and in our study. It seems that iron in the bone marrow of HD patients follows similar kinetics to that traditionally observed in secondary hemosiderosis.

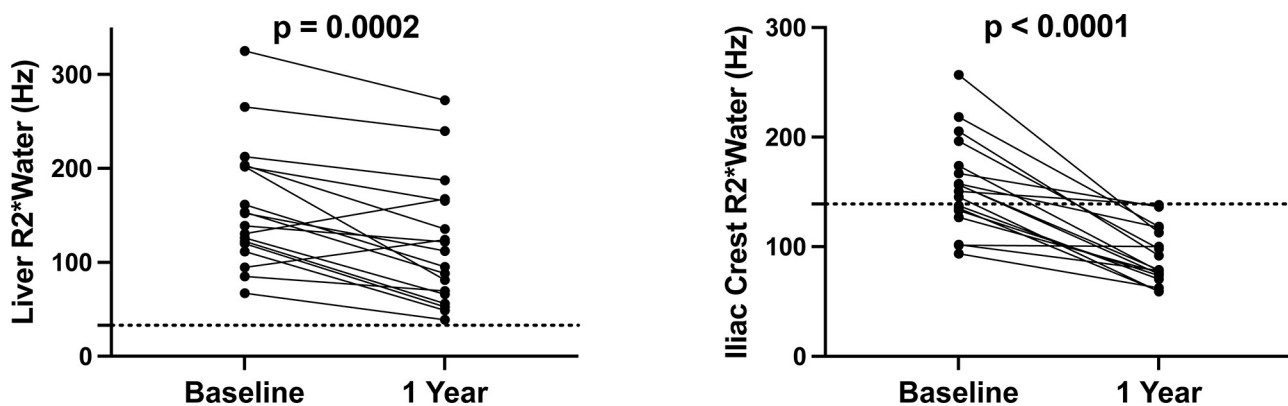

**Fig 4.** Liver R2*Water (A) and right iliac crest R2*Water (B) MRI parameter pre- and post-treatment.

**Table 3. Bone histomorphometry at baseline and at the end of follow-up.**

| | Baseline | 12 months | Absolute change | p |
|---|---|---|---|---|
| BV/TV (%) | 18.96 (14.3, 26.4) | 17.3 (12.3, 21.3) | -2.67 ± 1.90 | 0.349 |
| Tb.Th (μm) | 117.4 ± 21.8 | 122.8 ± 24.4 | 5.4 ± 3.87 | 0.486 |
| Tb.Sp (μm) | 503.3 (351, 613) | 590.2 (483, 775.2) | 184.6 ± 78.4 | **0.039** |
| Tb.N (μm) | 1.51 (1.33, 1.92) | 1.39 (1.14, 1.71) | -0.32 ± 1.79 | 0.093 |
| Fb.V/TV (%) | 0.085 (0.008, 0.548) | 0.045 (0.008, 0.505) | -2.96 ± 0.67 | 0.196 |
| OV/BV (%) | 6.47 (1.38, 13.8) | 4.59 (1.75, 12.0) | 0.79 ± 2.29 | 0.744 |
| O.Th (μm) | 11.2 (5.6, 13.2) | 10.0 (7.4, 13.8) | 2.88 ± 1.36 | 0.248 |
| OS/BS (%) | 40.4 ± 25.6 | 38.1 ± 24.5 | -2.3 ± 5.9 | 0.70 |
| Ob.S/BS (%) | 11.4 (4.4, 19.5) | 7.6 (3.2, 13.4) | -5.5 ± 2.4 | **0.043** |
| ES/BS (%) | 8.3 ± 4.6 | 7.0 ± 5.1 | -0.7 ± 1.3 | 0.348 |
| Oc.S/Bs (%) | 0.98 (0.37, 2.06) | 0.82 (0.31, 1.44) | 0.06 ± 0.39 | 0.679 |
| MAR (μm/d) | 1.01 ± 0.37 | 0.89 ± 0.50 | 0.12 ± 0.18 | 0.705 |
| MS/BS (%) | 8.21 (2.56, 9.92) | 5.46 (2.67, 6.82) | -2.22 ± 1.78 | 0.272 |
| BFR/BS ($μm^3/μm^2/d$) | 0.083(0.015, 0.13) | 0.036 (0.017, 0.073) | -0.03 ± 0.01 | 0.068 |
| Mlt (d) | 59 (38;94) | 52 (35.5, 128.8) | 9.5 ± 29.2 | 0.610 |
| CT.Th (μm) | 677 (528;830) | 675 (557, 774) | -32.1 ± 82.1 | 0.943 |
| Ct.Po (%) | 9.5 (7, 17) | 9.6 (6.4, 12.8) | -1.9 ± 1.4 | 0.523 |
| N.Cells Fe + (n) | 5057 (2228, 7044) | 332 (50, 860) | -4724 ± 764 | **<0.001** |
| Cells Fe+/Ma.Ar ($n/mm^2$) | 203 (124.6;287.6) | 9.51 (1.35;26.31) | -185.1 ± 24.0 | **<0.001** |

Values are expressed as the mean and standard deviation or median (25,75) for variables at baseline and 12 months and mean ± SEM for absolute change; BV/TV: bone volume; Tb.Th: trabecular thickness; Tb.Sp: trabecular separation; Tb.N: trabecular number; OV/BV: osteoid volume; O.Th: osteoid thickness; OS/BS: osteoid surface; Ob.S/BS: osteoblastic surface; ES/BS: resorption surface; Oc.S/BS: osteoclastic surface; Fb.V/TV: fibrosis volume; MS/BS: mineralizing surface; MAR: mineral apposition rate; BFR/BS: bone formation rate; Mlt: mineralization lag time; CT.Th: cortical thickness; Ct.Po (%): cortical porosity; Cells Fe+: Bone marrow cells stained for iron per tissue area; Cells Fe+/Ma.Ar

However, even in light of this evidence, the KDIGO guidelines continue to propose serum ferritin levels of 500 ng/L and transferrin saturation up to 30% [4], in agreement with the European Renal Best Practice [24]. In contrast, the Canadian Society of Nephrology [25] suggests lower TSAT and ferritin thresholds. The Japanese Society for Dialysis Therapy [26]

**Table 4. Turnover, Mineralization, and Volume (TMV) classification at baseline and 12 months after treatment with DFO.**

| TMV classification | Baseline | 12 months | p value |
|---|---|---|---|
| **T (turnover)** | | | |
| High | 12 (66.7%) | 12 (66.7%) | |
| Low | 6 (33.3%) | 4 (22.2%) | 0.30 |
| Normal | 0 (0%) | 2 (11.1%) | |
| **M (mineralization)** | | | |
| Normal | 6 (33.3%) | 7 (38.9%) | |
| Abnormal | 12 (66.7%) | 11 (61.1%) | 1 |
| **V (volume)** | | | |
| Normal | 12 (66.7%) | 12 (66.7%) | |
| Low | 5 (27.8%) | 5 (27.8%) | 1 |
| High | 1 (5.5%) | 1 (5.5%) | |

Data are expressed as frequency (percentage); Chi-Square test and Fisher's exact test

**Fig 5. Turnover, Mineralization, and Volume (TMV) classification at the different timepoints.** TMV distribution and individual evolution at baseline and at the 12-month follow-up.

establishes that iron supplementation is indicated only if the serum ferritin level is <100 ng/mL and the transferrin saturation rate (TSAT) is < 20%. Recently, a new piece of evidence has been added to the literature in favor of Fe supplementation. In the PIVOTAL study [27], patients on HD who received a median monthly dose of 264 mg had fewer cardiovascular events than those who received 145 mg. However, it should be noted that the PIVOTAL study included patients with < 12 months of therapy, without pronounced inflammation status, and from a single country. Therefore, the results may not be generalizable. In summary, while some studies advocate higher iron supplementation, others strongly recommend against it, fearing intoxication [28].

We found no adverse effects when withdrawing Fe supplementation and administering DFO. Since 2013, there have been studies describing the use of the iron chelator Deferasirox, (used orally) in both patients treated with HD or PD [29]. These studies evaluated a small number of patients and many of them were case reports. However, it is worth highlighting that this drug represents a promising option for the treatment of patients with CKD and iron overload [30–32]. However, larger studies are needed to demonstrate its safety and efficacy, especially in HD patients.

Hemoglobin levels remained virtually the same, and there was no need to increase the erythropoietin dose. Although it is well known that inflammation leads to increased serum ferritin [33] one cannot rule out Fe overload diagnosis in this scenario. Indeed, 50% of our patients had high CRP levels, but all had liver and bone marrow iron accumulation. The subgroup of patients with high serum ferritin levels had persistent Fe overload at the end of the follow-up. Therefore, Fe tissue overload should not be imputed to inflammation, at least for patients with extremely high serum ferritin levels. Sensitive inflammatory markers such as FGF-23 and hepcidin decreased during follow-up, indicating that excess Fe might have contributed to the inflammatory status [34], which improved after treatment, suggesting lower resistance to erythropoietin.

Liver MRI is the best noninvasive method for quantifying hepatic Fe levels. This technique has good sensitivity and specificity in both diagnosis and follow-up treatment of assorted pathologies [35, 36]. In the current study, the use of the R2*Water MRI parameter reduced the chance of bias due to fat deposits, thereby ameliorating the diagnostic accuracy of Fe overload. It is already known that liver Fe accumulation correlated with serum ferritin [2, 37, 38]. We showed this correlation even in individuals with serum ferritin levels as high as > 1000ng/ml, in the bone marrow and liver. Fe overload has been described to be associated with hepatic steatosis [39, 40].

In thalassemic patients, cardiac complications are responsible for about 50% of deaths, and the use of MRI for diagnosis led to an intensification of Fe chelation regimens with significant improvement in survival [41]. No evidence of cardiac iron overload was found. It seems that Fe deposition kinetics in the heart is slower than that in the liver, probably requiring a much longer time of Fe exposure to impregnate the cardiac tissue [41, 42].

Patients with CKD may have other heart MRI changes that are not associated with iron supplementation. However, it is noteworthy that excess Fe increases hepcidin levels, which in turn activates macrophages in atheromatous plaques, favoring their rupture. Thus, long-term excess Fe without correction would contribute to the cardiovascular complications of these patients [43, 44] A recent review stated that intravenous Fe therapy has been associated with an increased risk of atherothrombosis, vascular calcification, oxidative stress, and infection [45].

In the bone biopsy evaluation, we found that all patients had iron overload in the bone marrow. Rocha et al. [8] described a quantitative method for the counting of iron-stained cells in the bone marrow of an HD patient and observed an increase in cells in patients with ferritin levels above 500 ng/ml. To the best of our knowledge, the current study is the first to evaluate Fe deposits in the iliac crests using MRI. Despite the lack of reference values for this region, the significant correlations between the amount of liver Fe values and the lumbar spine and iliac crest suggest that the iliac crest could also reflect the excess of Fe in the body. Another interesting result is that serum ferritin levels are a marker of Fe tissue accumulation because they correlate with T2*, R2* and R2*Water values in the liver and with R2*Water in the iliac crests. Although this method has already been validated in patients with Fe overload [41–46] this is the first time it has been used in patients with CKD.

Fe is involved in bone formation, iron deficiency and iron-deficiency anemia are linked to phosphate metabolism, and FGF-23 transcription is elevated in iron deficiency. Cell line studies of immortalized human fetal osteoblasts (hFOB1.19) have shown that Fe increases reactive oxygen species, decreases alkaline phosphatase activity, and impairs bone mineralization. Other studies have demonstrated that Fe decreases the expression of osteocalcin and RUNX2 (runt-related transcription factor 2) which affect bone formation [47–50]. Therefore, one could expect that withdrawal of Fe supplementation and DFO administration would change BMD biomarkers and bone histomorphometry. However, our results did not meet these expectations. The lack of studies on patients with advanced CKD precluded a comparison of the results.

In recent years, the discovery of Fe-based P binders has been proposed as a promising alternative to both offer Fe and chelate P [51]. However, there are no long-term evaluation studies on whether the supply of Fe would lead to overload and its effect on mineral metabolism disorders. Regarding the histomorphometry analysis of bone biopsy, we found that remodelling and mineralization were compromised in most patients, and bone volume decreased in 35% of patients. These findings were observed in cohorts of patients with CKD who underwent bone biopsy and were apparently not influenced by Fe overload [52, 53]. Prospective studies including patients with CKD and Fe overload treated with Fe binders and bone biopsy before and after treatment might shed light on the role of Fe in the observed bone changes.

Our study has some limitations. First, the sample size is small. Second, we assessed a specific study population characterized by a serum ferritin level > 1000 ng/ml and demonstrated a direct relationship between ferritin and Fe tissue accumulation. Our results may not be generalizable to all patients on dialysis because of the inclusion of a specific group of patients. These limitations are balanced against several strengths, including the prospective design and use of MRI, which is a sensitive method for assessing Fe overload. In addition, it includes multiple techniques to test the relationship and agreement among them. Finally, it highlights the novelty of applying iliac crest MRI for the assessment of Fe accumulation.

In conclusion, this study showed a direct association between high ferritin levels and liver and bone marrow Fe overload in patients with CKD undergoing maintenance hemodialysis. The withdrawal of Fe supplementation and DFO administration was safe and ameliorated the Fe overload.

## Supporting information

**S1 Checklist. TREND statement checklist.**
(DOCX)

**S1 File. Trial study protocol.**
(PDF)

**S2 File. Methods.**
(PDF)

**S1 Table. Characteristics of patients.**
(PDF)

**S2 Table. Bone histomorphometric parameters.**
(PDF)

**S3 Table. Magnetic resonance imaging.**
(PDF)

## Acknowledgments

The authors thank Rosimeire A. B. Costa and Wagner V. Dominguez for technical support.

## Author Contributions

**Conceptualization:** Lucas L. A. Nunes, Hilton Leão Filho, Vanda Jorgetti, Melani R. Custodio.

**Data curation:** Lucas L. A. Nunes, Luciene M. Dos Reis, Hanna K. A. Guapyassú, Rosilene M. Elias, Vanda Jorgetti, Melani R. Custodio.

**Formal analysis:** Lucas L. A. Nunes, Luciene M. Dos Reis, Rosse Osorio, Hanna K. A. Guapyassú, Rosa M. A. Moysés, Hilton Leão Filho, Rosilene M. Elias, Carlos E. Rochitte, Vanda Jorgetti, Melani R. Custodio.

**Funding acquisition:** Rosse Osorio, Hanna K. A. Guapyassú.

**Investigation:** Lucas L. A. Nunes, Rosse Osorio, Hanna K. A. Guapyassú.

**Methodology:** Luciene M. Dos Reis, Rosse Osorio, Rosa M. A. Moysés, Hilton Leão Filho, Rosilene M. Elias, Carlos E. Rochitte, Vanda Jorgetti, Melani R. Custodio.

**Project administration:** Rosa M. A. Moysés, Vanda Jorgetti, Melani R. Custodio.

**Resources:** Vanda Jorgetti, Melani R. Custodio.

**Software:** Lucas L. A. Nunes, Rosse Osorio, Rosa M. A. Moysés, Hilton Leão Filho, Rosilene M. Elias, Carlos E. Rochitte.

**Supervision:** Luciene M. Dos Reis, Vanda Jorgetti, Melani R. Custodio.

**Validation:** Luciene M. Dos Reis, Rosilene M. Elias, Carlos E. Rochitte, Vanda Jorgetti, Melani R. Custodio.

**Visualization:** Rosa M. A. Moysés, Vanda Jorgetti, Melani R. Custodio.

**Writing – original draft:** Lucas L. A. Nunes, Hilton Leão Filho.

**Writing – review & editing:** Lucas L. A. Nunes, Rosa M. A. Moysés, Vanda Jorgetti, Melani R. Custodio.

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
