## [Decision Letter · Decision Letter 0]

15 Mar 2024

PONE-D-24-04153High ferritin is associated with liver and bone marrow iron accumulation: effects of 1-year deferoxamine treatmentPLOS ONE

Dear Dr. Nunes,

Thank you for submitting your manuscript to PLOS ONE. After careful consideration, we feel that it has merit but does not fully meet PLOS ONE’s publication criteria as it currently stands. Therefore, we invite you to submit a revised version of the manuscript that addresses the points raised during the review process.

We look forward to receiving your revised manuscript.

Kind regards,

Aleksandra Klisic

Academic Editor

PLOS ONE

Journal Requirements:

"This work was supported by the Brazilian National Council for Scientific and

Technological Development (CNPq; grant number 434758/2018-3)."

"All the authors declared no competing interests"

6. We note that Figure 3 in your submission contain copyrighted images. All PLOS content is published under the Creative Commons Attribution License (CC BY 4.0), which means that the manuscript, images, and Supporting Information files will be freely available online, and any third party is permitted to access, download, copy, distribute, and use these materials in any way, even commercially, with proper attribution. For more information, see our copyright guidelines: http://journals.plos.org/plosone/s/licenses-and-copyright.

a. You may seek permission from the original copyright holder of Figure 3 to publish the content specifically under the CC BY 4.0 license. 

7. We note that you have selected “Clinical Trial” as your article type. PLOS ONE requires that all clinical trials are registered in an appropriate registry (the WHO list of approved registries is at      https://www.who.int/clinical-trials-registry-platform/network/primary-registries"" https://www.who.int/clinical-trials-registry-platform/network/primary-registries" https://www.who.int/clinical-trials-registry-platform/network/primary-registries and more information on trial registration is at http://www.icmje.org/about-icmje/faqs/clinical-trials-registration/). 

Please state the name of the registry and the registration number (e.g. ISRCTN or ClinicalTrials.gov) in the submission data and on the title page of your manuscript.

a) Please provide the complete date range for participant recruitment and follow-up in the methods section of your manuscript.

b) If you have not yet registered your trial in an appropriate registry, we now require you to do so and will need confirmation of the trial registry number before we can pass your paper to the next stage of review. Please include in the Methods section of your paper your reasons for not registering this study before enrolment of participants started. Please confirm that all related trials are registered by stating: “The authors confirm that all ongoing and related trials for this drug/intervention are registered”.

Please see http://journals.plos.org/plosone/s/submission-guidelines#loc-clinical-trials for our policies on clinical trials.

Reviewers' comments:

Reviewer's Responses to Questions

**Comments to the Author**

1. Is the manuscript technically sound, and do the data support the conclusions?

Reviewer #1: Yes

Reviewer #2: Partly

2. Has the statistical analysis been performed appropriately and rigorously? 

Reviewer #1: Yes

Reviewer #2: I Don't Know

3. Have the authors made all data underlying the findings in their manuscript fully available?

Reviewer #1: Yes

Reviewer #2: Yes

4. Is the manuscript presented in an intelligible fashion and written in standard English?

Reviewer #1: Yes

Reviewer #2: Yes

5. Review Comments to the Author

Reviewer #1: In this very high-quality article, the authors analyze prospectively with R2* relaxometry both liver, heart, spine bone marrow and iliac bone marrow iron content together with bone marrow biopsy in a cohort of hemodialysis patients highly iron overloaded treated with Deferoxamine, an IV classical iron chelator given at the end of dialysis sessions. They elegantly show that heart was speared in these patients and that iron deposits are significantly reduced in liver and bone barrow (at spine and crest iliac sites). This notion of benefit of iron chelation by Deferoxamine in dialysis associated iron overload in the modern ESA era appears clearly new with an important clinical translation.

I suggest some modifications to allow this article to reach the high standard of publications of Plos one:

-Title and running title :

Title :the readers do not imagine that the study focus on dialysis patients ; it seems wise to add the term dialysis as follows : High ferritin is associated with liver and bone marrow iron accumulation : effects of 1- year deferoxamine treatment “in hemodialysis associated iron overload”

Running Title : Liver and bone marrow iron accumulation “in dialysis”

-Line 103-104 : Since most of these studies have been performed before EPO discovery, I would add to the original sentence : “However, few studies have reported the use of DFO to treat iron overload in patients with CKD, mostly in the pre-ESA era”13-15

-Line 132-142 : Normal values of the different parameters are best given in the tables (left column) instead of the text.

-Pages 7 and 8: did the authors find a correlation between LIC and monthly dose of IV iron; what were the correlation between spine R2*water and Liver R2* water, and between ferritin and spine R2*water ?

-Table 2, page 10 : For liver iron concentration given as R2* and T2* , authors also give LIC values in mgFe /g of dry weight; they also should give values of LIC in micromole Fe /g of dry weight (largely used) to allow numerous clinicians to easily understand the results. Similarly, for iliac and spine iron, iron concentration values are given as R2*values , but authors should also give values in T2* values (ms) which are largely used in the literature.

-The discussion section should include others relevant publications on this topic:

1) Carrilho P, Fidalgo P, Lima A, et al. Post-mortem liver and bone marrow iron quantification in haemodialysis patients: a prospective cohort study. EBioMedicine. 2022;77:103921. doi: 10.1016/j. 25. Ali M, Fayemi AO, Rigolosi R, Frascino J, Marsden T, Malco ebiom.2022.103921. This post-mortem study comprising immediate post-mortem specimen in 21 hemodialysis Portugese patients showed the parallelism between the bone marrow and liver compartments.

2) Rostoker G, Dekeyser M, Francisco S, et al. Relationship between bone marrow iron load and liver iron concentration in dialysis-associated haemosiderosis. EBioMedicine. 2024 Jan;99:104929. doi:10.1016/j.ebiom.2023.104929. Epub 2023 Dec 20.

This MRI-R2* relaxometry study (of liver, spleen and spine) recently published confirms and extends, in a cohort of 152 alive patients treated by hemodialysis with iatrogenic iron overload in about one half of them, the conclusions of the recent autopsy study by Carrilho et al. Both studies demonstrate that the paradoxical discrepancy observed by Ali et al. in the pre-ESA era (Ali M, Rigolosi R, Fayemi AO,et al. Failure of serum ferritin levels to predict bone-marrow iron content after intravenous iron-dextran therapy. Lancet. 1982;1(8273):652-655. doi: 10.1016/S0140-6736(82)92204-8.) between the bone marrow and hepatic iron compartments in overloaded dialysis patients has totally disappeared in the current modern ESA era, and that bone marrow iron now rather follows a similar kinetic, as traditionally seen in secondary hemosiderosis.

3) At least 3 publications have documented the use of the oral chelator Deferasirox (probably less safe than Deferoxamine) in dialysis patients suffering from iron overload (article 3-1 having MRI monitoring)

3-1) Bekhechi W, Chiali H, Khelil L, Sari-Hamidou R, Benmansour M. Hemosiderosis in chronic dialysis patients: Monitoring the response to deferasirox by quantitative hepatic magnetic resonance imaging. Hemodial Int. 2023 Jul;27(3):270-277. doi: 10.1111/hdi.13081. Epub 2023 Mar 30.

3-2) Yii E, Doery JC, Kaplan Z, Kerr PG. Use of deferasirox (Exjade) for iron overload in peritoneal dialysis patients.Nephrology (Carlton). 2018 Sep;23(9):887-889. doi: 10.1111/nep.13389.

3-3) Chen CH, Shu KH, Yang Y Long-term effects of an oral iron chelator, deferasirox, in hemodialysis patients with iron overload..Hematology. 2015 Jun;20(5):304-10. doi: 10.1179/1607845414Y.0000000199. Epub 2014 Sep 9.PMID: 25200910

Reviewer #2: This is an interesting paper on the use of DFO in patients with high ferritin and chronic kidney disease. The analyses seem to be largely well-performed, but there is a lack of information on the design of the study.

In terms of design it would help to

a) explain why 28 patients was considered sufficient here - was there a power calculation done, and what was the primary outcome in the study? The flowchart should be introduced in line 157 and the inconsistency between 70 and 95 as the starting point explained here.

b) In terms of time of study, it makes sense to separate recruitment and follow-up - it would appear to be 1 year recruitment and then 1 year follow-up but this is not very clear.

When analysing the data, they are of course paired, so it would help to give the change score distribution here with mean and sem or equivalent as this is what drives the p-value and it helps to understand the scale of the difference as this is probably more important than the significance level from a clinical point of view.

Figure 5 is more precise as a table as numbers are not given here. A table can also give the extent of the data as paired rates for McNemars test etc.

6. PLOS authors have the option to publish the peer review history of their article (what does this mean?). If published, this will include your full peer review and any attached files.

Reviewer #1: No

Reviewer #2: No

---

## [Author Response · Author response to Decision Letter 0]

6 May 2024

Response to Academic Editor 

* 1. Please ensure that your manuscript meets PLOS ONE's style requirements, including those for file naming. The PLOS ONE style templates can be found at https://journals.plos.org/plosone/s/file?id=wjVg/PLOSOne_formatting_smple_main_body.pdf and https://journals.plos.org/plosone/s/file?id=ba62/PLOSOne_formatting_sample_title_authors_affiliations.pdf

R: Thank you for your observation. We agree with the academic editor and have changed the manuscript.

*2. We note that the grant information you provided in the ‘Funding Information’ and ‘Financial Disclosure’ sections do not match. When you resubmit, please ensure that you provide the correct grant numbers for the awards you received for your study in the ‘Funding Information’ section.

R: We have corrected and matched the information.

*3. Thank you for stating the following financial disclosure: 

"This work was supported by the Brazilian National Council for Scientific and

Technological Development (CNPq; grant number 434758/2018-3)."

R: Thank you for your observation. We included the sentence in our cover letter, as suggested. 

*4. Thank you for stating the following in your Competing Interests section: "All the authors declared no competing interests"

R: Thank you for your observation. We have changed accordingly.

*5. We note that you have included the phrase “data not shown” in your manuscript. Unfortunately, this does not meet our data sharing requirements. PLOS does not permit references to inaccessible data. We require that authors provide all relevant data within the paper, Supporting Information files, or in an acceptable, public repository. Please add a citation to support this phrase or upload the data that corresponds with these findings to a stable repository (such as Figshare or Dryad) and provide and URLs, DOIs, or accession numbers that may be used to access these data. Or, if the data are not a core part of the research being presented in your study, we ask that you remove the phrase that refers to these data.

R: Thank you for your observation. We agree and have modified the manuscript in lines 173-177.

*6. We note that Figure 3 in your submission contain copyrighted images. All PLOS content is published under the Creative Commons Attribution License (CC BY 4.0), which means that the manuscript, images, and Supporting Information files will be freely available online, and any third party is permitted to access, download, copy, distribute, and use these materials in any way, even commercially, with proper attribution. For more information, see our copyright guidelines: http://journals.plos.org/plosone/s/licenses-and-copyright.

a. You may seek permission from the original copyright holder of Figure 3 to publish the content specifically under the CC BY 4.0 license. 

We recommend that you contact the original copyright holder with the Content Permission Form (http://journals.plos.org/plosone/s/file?id=7c09/content-permission-form.pdf) and the following text: “I request permission for the open-access journal PLOS ONE to publish XXX under the Creative Commons Attribution License (CCAL) CC BY 4.0 (http://creativecommons.org/licenses/by/4.0/). Please be aware that this license allows unrestricted use and distribution, even commercially, by third parties. Please reply and provide explicit written permission to publish XXX under a CC BY license and complete the attached form.”

Please upload the completed Content Permission Form or other proof of granted permissions as an "Other" file with your submission. In the figure caption of the copyrighted figure, please include the following text: “Reprinted from [ref] under a CC BY license, with permission from [name of publisher], original copyright [original copyright year].”

R: Thank you for your comment. We have uploaded the updated Figure as required.

*7. We note that you have selected “Clinical Trial” as your article type. PLOS ONE requires that all clinical trials are registered in an appropriate registry (the WHO list of approved registries is at https://www.who.int/clinical-trials-registry-platform/network/primary-registries" https://www.who.int/clinical-trials-registry-platform/network/primary-registries and more information on trial registration is at http://www.icmje.org/about-icmje/faqs/clinical-trials-registration/). 

Please state the name of the registry and the registration number (e.g. ISRCTN or ClinicalTrials.gov) in the submission data and on the title page of your manuscript.

a) Please provide the complete date range for participant recruitment and follow-up in the methods section of your manuscript.

b) If you have not yet registered your trial in an appropriate registry, we now require you to do so and will need confirmation of the trial registry number before we can pass your paper to the next stage of review. Please include in the Methods section of your paper your reasons for not registering this study before enrolment of participants started. Please confirm that all related trials are registered by stating: “The authors confirm that all ongoing and related trials for this drug/intervention are registered”.

R: Thank you for your observation. This study was registered at the REBEC (Registro Brasileiro de Ensaios Clinicos) at the website https://ensaiosclinicos.gov.br/welcome under the identification number #RBR-3rnskcj. This registry is part of the WHO (world Health Organization). (Recruitment - Feb 2017 to Feb 2019 and Follow-up - 1 year). 

Responses to the Reviewer 1

*Title and running title: Title :the readers do not imagine that the study focus on dialysis patients ; it seems wise to add the term dialysis as follows : High ferritin is associated with liver and bone marrow iron accumulation : effects of 1- year deferoxamine treatment “in hemodialysis associated iron overload”. Running Title : Liver and bone marrow iron accumulation “in dialysis”.

R.: Thank you for your observation. We have modified the title and running title. 

* Line 103-104 : Since most of these studies have been performed before EPO discovery, I would add to the original sentence : “However, few studies have reported the use of DFO to treat iron overload in patients with CKD, mostly in the pre-ESA era”13-15

R. We agree and thank you for your suggestion.

* Line 132-142: Normal values of the different parameters are best given in the tables (left column) instead of the text.

R. Thank you for your suggestion. We agree and have included this information in table 1.

* Pages 7 and 8: did the authors find a correlation between LIC and monthly dose of IV iron; what were the correlation between spine R2*water and Liver R2* water, and between ferritin and spine R2*water ?

R. Thank you for your suggestion. We have included this information in lines 168-170.

* Table 2, page 10: For liver iron concentration given as R2* and T2* , authors also give LIC values in mgFe /g of dry weight; they also should give values of LIC in micromole Fe /g of dry weight (largely used) to allow numerous clinicians to easily understand the results. Similarly, for iliac and spine iron, iron concentration values are given as R2*values , but authors should also give values in T2* values (ms) which are largely used in the literature.

R: Thank you for your suggestion. We have included this information in Table 2. 

* The discussion section should include others relevant publications on this topic: 

1) Carrilho P, Fidalgo P, Lima A, et al. Post-mortem liver and bone marrow iron quantification in haemodialysis patients: a prospective cohort study. EBioMedicine. 2022;77:103921. doi: 10.1016/j. 25. Ali M, Fayemi AO, Rigolosi R, Frascino J, Marsden T, Malco ebiom.2022.103921. This post-mortem study comprising immediate post-mortem specimen in 21 hemodialysis Portugese patients showed the parallelism between the bone marrow and liver compartments.

2) Rostoker G, Dekeyser M, Francisco S, et al. Relationship between bone marrow iron load and liver iron concentration in dialysis-associated haemosiderosis. EBioMedicine. 2024 Jan;99:104929. doi:10.1016/j.ebiom.2023.104929. Epub 2023 Dec 20. 

This MRI-R2* relaxometry study (of liver, spleen and spine) recently published confirms and extends, in a cohort of 152 alive patients treated by hemodialysis with iatrogenic iron overload in about one half of them, the conclusions of the recent autopsy study by Carrilho et al. Both studies demonstrate that the paradoxical discrepancy observed by Ali et al. in the pre-ESA era (Ali M, Rigolosi R, Fayemi AO,et al. Failure of serum ferritin levels to predict bone-marrow iron content after intravenous iron-dextran therapy. Lancet. 1982;1(8273):652-655. doi: 10.1016/S0140-6736(82)92204-8.) between the bone marrow and hepatic iron compartments in overloaded dialysis patients has totally disappeared in the current modern ESA era, and that bone marrow iron now rather follows a similar kinetic, as traditionally seen in secondary hemosiderosis.

3) At least 3 publications have documented the use of the oral chelator Deferasirox (probably less safe than Deferoxamine) in dialysis patients suffering from iron overload (article 3-1 having MRI monitoring)

3-1) Bekhechi W, Chiali H, Khelil L, Sari-Hamidou R, Benmansour M. Hemosiderosis in chronic dialysis patients: Monitoring the response to deferasirox by quantitative hepatic magnetic resonance imaging. Hemodial Int. 2023 Jul;27(3):270-277. doi: 10.1111/hdi.13081. Epub 2023 Mar 30.3-2) Yii E, Doery JC, Kaplan Z, Kerr PG. Use of deferasirox (Exjade) for iron overload in peritoneal dialysis patients.Nephrology (Carlton). 2018 Sep;23(9):887-889. doi: 10.1111/nep.13389.

3-3) Chen CH, Shu KH, Yang Y Long-term effects of an oral iron chelator, deferasirox, in hemodialysis patients with iron overload..Hematology. 2015 Jun;20(5):304-10. doi: 10.1179/1607845414Y.0000000199. Epub 2014 Sep 9.PMID: 25200910.

R. Thank you for your suggestion. We have added these new references in the discussion section. 

Responses to the Reviewer 2

*a) Explain why 28 patients was considered sufficient here - was there a power calculation done, and what was the primary outcome in the study? The flowchart should be introduced in line 157 and the inconsistency between 70 and 95 as the starting point explained here.

R. a) The sample size was obtained by convenience since there was no previous study that had evaluated R2*Water as a measurement for iron organ deposition. All patients who fulfilled the inclusion criteria were invited to participate. We have made this clear in the updated version of the manuscript, as shown in the flowchart. 

*b) In terms of time of study, it makes sense to separate recruitment and follow-up - it would appear to be 1 year recruitment and then 1 year follow-up but this is not very clear.

R.b): Thank you for your observation. We have changed it in the text.

*When analysing the data, they are of course paired, so it would help to give the change score distribution here with mean and sem or equivalent as this is what drives the p-value and it helps to understand the scale of the difference as this is probably more important than the significance level from a clinical point of view.

R: Thank you for your suggestion. We agree and have included this information in Tables 1,2 and 3. 

*Figure 5 is more precise as a table as numbers are not given here. A table can also give the extent of the data as paired rates for McNemars test etc.

R. Thank you for your suggestion. We agree and have included this information in Table 4.

---

## [Decision Letter · Decision Letter 1]

24 May 2024

PONE-D-24-04153R1High ferritin is associated with liver and bone marrow iron accumulation: effects of 1-year deferoxamine treatment in hemodialysis-associated iron overloadPLOS ONE

Dear Dr. Nunes,

Thank you for submitting your manuscript to PLOS ONE. After careful consideration, we feel that it has merit but does not fully meet PLOS ONE’s publication criteria as it currently stands. Therefore, we invite you to submit a revised version of the manuscript that addresses the points raised during the review process.

We look forward to receiving your revised manuscript.

Kind regards,

Aleksandra Klisic

Academic Editor

PLOS ONE

Journal Requirements:

Reviewers' comments:

Reviewer's Responses to Questions

**Comments to the Author**

1. If the authors have adequately addressed your comments raised in a previous round of review and you feel that this manuscript is now acceptable for publication, you may indicate that here to bypass the “Comments to the Author” section, enter your conflict of interest statement in the “Confidential to Editor” section, and submit your "Accept" recommendation.

Reviewer #1: All comments have been addressed

Reviewer #2: (No Response)

2. Is the manuscript technically sound, and do the data support the conclusions?

Reviewer #1: Yes

Reviewer #2: Yes

3. Has the statistical analysis been performed appropriately and rigorously? 

Reviewer #1: Yes

Reviewer #2: Yes

4. Have the authors made all data underlying the findings in their manuscript fully available?

Reviewer #1: Yes

Reviewer #2: Yes

5. Is the manuscript presented in an intelligible fashion and written in standard English?

Reviewer #1: Yes

Reviewer #2: Yes

6. Review Comments to the Author

**Reviewer #1:** This revised version has considered reviewers’ comments and advice ad seems suitable for publication in PLOS ONE. One minor remark: authors please explain the acronym TMV.

**Reviewer #2:** Thank you for your previous responses. I would request one chance to Table 4 - this is presented as if the data are not paired - please instead cross tabulate the values at the different timepoints- the data currently presented is therefore in the marginal data, but one can see the paired data allowing clarity of testing.

7. PLOS authors have the option to publish the peer review history of their article (what does this mean?). If published, this will include your full peer review and any attached files.

Reviewer #1: No

Reviewer #2: No

---

## [Author Response · Author response to Decision Letter 1]

4 Jun 2024

Response to Academic Editor 

* 1. Please review your reference list to ensure that it is complete and correct.

R: Thank you for your observation. We have review and included the correct references in line 148 and 202.

Responses to the Reviewer 1

* This revised version has considered reviewers’ comments and advice ad seems suitable for publication in PLOS ONE. One minor remark: authors please explain the acronym TMV.

R.: Thank you for your observation. We have explained the acronym TMV in line 185-187. 

Responses to the Reviewer 2

* Thank you for your previous responses. I would request one chance to Table 4 - this is presented as if the data are not paired - please instead cross tabulate the values at the different timepoints- the data currently presented is therefore in the marginal data, but one can see the paired data allowing clarity of testing.

R: Thank you for your suggestion. We agree and have included this information in Figure 5

---

## [Decision Letter · Decision Letter 2]

14 Jun 2024

High ferritin is associated with liver and bone marrow iron accumulation: effects of 1-year deferoxamine treatment in hemodialysis-associated iron overload

PONE-D-24-04153R2

Dear Dr. Nunes,

We’re pleased to inform you that your manuscript has been judged scientifically suitable for publication and will be formally accepted for publication once it meets all outstanding technical requirements.

Kind regards,

Aleksandra Klisic

Academic Editor

PLOS ONE

Additional Editor Comments (optional):

Reviewers' comments:

Reviewer's Responses to Questions

**Comments to the Author**

1. If the authors have adequately addressed your comments raised in a previous round of review and you feel that this manuscript is now acceptable for publication, you may indicate that here to bypass the “Comments to the Author” section, enter your conflict of interest statement in the “Confidential to Editor” section, and submit your "Accept" recommendation.

Reviewer #2: All comments have been addressed

2. Is the manuscript technically sound, and do the data support the conclusions?

Reviewer #2: (No Response)

3. Has the statistical analysis been performed appropriately and rigorously? 

Reviewer #2: (No Response)

4. Have the authors made all data underlying the findings in their manuscript fully available?

Reviewer #2: (No Response)

5. Is the manuscript presented in an intelligible fashion and written in standard English?

Reviewer #2: (No Response)

6. Review Comments to the Author

Reviewer #2: (No Response)

7. PLOS authors have the option to publish the peer review history of their article (what does this mean?). If published, this will include your full peer review and any attached files.

Reviewer #2: No

---

## [Editor Report · Acceptance letter]

1 Jul 2024

PONE-D-24-04153R2 

PLOS ONE

Dear Dr. Nunes, 

I'm pleased to inform you that your manuscript has been deemed suitable for publication in PLOS ONE. Congratulations! Your manuscript is now being handed over to our production team.

Kind regards, 

on behalf of

Dr. Aleksandra Klisic 

Academic Editor

PLOS ONE